# Synthesis of Water-Dispersed Sulfobetaine Methacrylate–Iron Oxide Nanoparticle-Coated Graphene Composite by Free Radical Polymerization

**DOI:** 10.3390/polym14183885

**Published:** 2022-09-17

**Authors:** Suguna Perumal, Raji Atchudan, Yong Rok Lee

**Affiliations:** 1Department of Chemistry, Sejong University, Seoul 143-747, Korea; 2School of Chemical Engineering, Yeungnam University, Gyeongsan 38541, Korea; 3Department of Chemistry, Saveetha School of Engineering, Saveetha Institute of Medical and Technical Sciences, Chennai 602105, Tamil Nadu, India

**Keywords:** graphene composites, iron oxide nanoparticles, poly[2-(methacryloyloxy)ethyl]dimethyl-(3-sulfopropyl)ammonium hydroxide], sonication bath, thin-layered graphene

## Abstract

Research on the synthesis of water-soluble polymers has accelerated in recent years, as they are employed in many bio-applications. Herein, the synthesis of poly[2-(methacryloyloxy)ethyl]dimethyl-(3-sulfopropyl)ammonium hydroxide (PSB) by free radical polymerization in a sonication bath is described. PSB and iron oxide nanoparticles (IONPs) were simultaneously stabilized on the graphene surface. Graphene surfaces with PSB (GPSB) and graphene surfaces with PSB and IONPs (GPSBI) were prepared. Since PSB is a water-soluble polymer, the hydrophobic nature of graphene surfaces converts to hydrophilic nature. Subsequently, the prepared graphene composites, GPSB and GPSBI, were well-dispersed in water. The preparation of GPSB and GPSBI was confirmed by X-ray diffraction, Raman spectroscopy, field emission scanning electron microscopy, transmission electron microscopy, X-ray photoelectron spectroscopy, and thermogravimetric analysis. The impacts of PSB and IONPs on the graphene surfaces were studied systematically.

## 1. Introduction

Continuous research is ongoing towards the development of diagnosis and therapeutic agents for detecting and treating cancer cells. Nanostructures and hybrid nanostructures are used for diagnosis and therapeutic applications [1,2,3,4]. These nanostructures can be tuned to accommodate the desired properties. In recent times, iron oxide nanomaterials (IONPs) attained considerable attention in various fields, including biomedical, diagnostic, and therapeutic applications [5,6,7]. The magnetic surface and intrinsic properties such as colloidal stability, low toxicity, and uniform size allowed researchers to use IONPs in different applications [7,8,9]. However, compared to IONPs alone, composites, especially those with nanoscale dimensions, have improved properties and are used in various applications [10,11,12,13]. In composites, research on graphene with IONPs is emphasized because of their excellent properties [14,15].

Graphene is a two-dimensional material composed of sp^2^ carbon atoms [16,17]. Graphene is an excellent material that has superior mechanical, electrical, and thermal properties [18,19]. Because of these properties, graphene and graphene composites are used in broad applications such as supercapacitors, water treatment, and biomedical applications [20,21,22,23,24,25]. Size-controlled graphene sheets with IONPs [26] showed remarkable catalytic activity in oxygen reduction reaction (ORR) and oxygen evolution reaction (OER) [27]. Three-dimensional reduced graphene oxide surface with IONPs was reported as an effective active material for deionization electrodes [28]. Superparamagnetic IONPs with graphene oxide (GO) are used as a resonance contrast agent for magnetic resonance imaging [29]. GO with IONPs is suggested for resonance/fluorescence imaging and cancer sensing applications [29,30]. In addition, IONPs grafted on GO surfaces are used for hyperthermia applications [31]. The polymers on the graphene surface tune the properties of graphene composites [32,33,34,35,36]. Different types of biocompatible polymers are used to stabilize the graphene surface for cancer theranostics [37,38,39,40,41]. Recently, we published a study on graphene nanocarriers for treating thyroid cancer cells [42]. Doxorubicin-loaded 2-(methacryloxyloxy)ethyl phosphorylcholine and poly(ethylene glycol) monomethacrylate stabilized the graphene surface with IONPs, representing a remarkable nanocarrier [42].

In this work, [2-(methacryloyloxy)ethyl]dimethyl-(3-sulfopropyl)ammonium hydroxide (SB) monomer was polymerized on the graphene surface, and graphite (G) was exfoliated into thin-layered graphene sheets by sonication. Two composites were prepared in the absence and presence of IONPs, GPSB and GPSBI, respectively. The prepared GPSB and GPSBI composites were characterized using various studies.

## 2. Materials and Methods

### 2.1. Materials

SB (95%), G, and 4,4′-azobis(4-cyanovaleric acid) (ACVA, ≥98%) were purchased from Sigma-Aldrich, South Korea, and used as received. Deionized water (DI) was used in all experiments. Using a bath sonicator (40 kHz, 110 W, BRANSON 3800, Richmond VA, USA), in situ polymerization of SB monomer was performed on the graphene surface at 70 °C. The composites were centrifuged using VS-18000M, VISION Scientific Co., Ltd., Daejeon-Si, Korea.

### 2.2. Methods

PSB, GPSB, and GPSBI were characterized using various physicochemical techniques. Raman spectra for composites were obtained on the XploRA Micro-Raman spectrophotometer (Horiba) in the range between 1000 and 3000 cm^−1^. X-ray diffraction (XRD) studies were carried out using the PANalytical X’Pert^3^ MRD diffractometer with monochromatized Cu Kα radiation (λ = 1.54 Å) at 40 kV and 30 mA and were recorded in the range from 20° to 80° (2*θ*). Field emission scanning electron microscopy (FESEM) with energy-dispersive X-ray spectroscopy (EDS) was used to evaluate the surface morphology of the composites. Using the Hitachi S-4800 equipped with EDX at an accelerating voltage of 10 kV, FESEM and EDS measurements were carried out. Transmission electron microscopy (TEM) images were obtained from JEOL JEM with an operating accelerating voltage of 120 kV. X-ray photoelectron spectroscopy (XPS) spectra were recorded using K-Alpha (Thermo Scientific, Waltham, MA, USA), and CasaXPS software was used for the deconvolution of the high-resolution XPS spectra. Thermogravimetric analysis (TGA) measurements were carried out on SDT Q600 with nitrogen atmosphere over 0–900 °C with 10 °C/min.

### 2.3. Graphene-poly[2-(Methacryloyloxy)ethyl]dimethyl-(3-sulfopropyl)ammonium Hydroxide] Composite

The preparation of graphene-poly[2-(methacryloyloxy)ethyl]dimethyl-(3-sulfopropyl)ammonium hydroxide] composite (GPSB) was prepared as shown in Figure 1. Monomer SB (500 mg, 1.78 mmol), ACVA (25.0 mg, 0.089 mmol), and 250 mg of G in 70 mL of DI water were heated at 70 °C for 6 h in a sonication bath. Then, the composite GPSB was purified by centrifugation at 5000 rpm for 15 min. Triplicate centrifugation followed by drying in a freeze dryer yielded fine powder of GPSB that was re-dispersed in DI water for further characterization.

### 2.4. Graphene-poly[2-(Methacryloyloxy)ethyl]dimethyl-(3-sulfopropyl)ammonium Hydroxide]–Iron Oxide Nanoparticle Composite

The preparation of the graphene-poly[2-(methacryloyloxy)ethyl]dimethyl-(3-sulfopropyl)ammonium hydroxide]–iron oxide nanoparticle composite (GPSBI) was prepared as shown in Figure 2. To prepare GPSBI, IONPs were prepared following earlier reports [43,44]. Iron acetylacetonate (3.0 g, 8.49 mmol) in 60 mL of benzyl alcohol was added to an autoclave container and heated at 180 °C for 48 h. The precipitates were purified by washing with ethanol and then centrifuged. Successively the precipitates were further washed with dichloromethane and then centrifuged. The purified IONPs were dried in a hot-air oven at 60 °C and used for the preparation of GPSBI composite. Monomer SB (500 mg, 1.78 mmol), ACVA (25.0 mg, 0.089 mmol), 250 mg of G, and 50 mg of IONPs were added to 70 mL of DI water, and the mixture was heated at 70 °C for 6 h in a sonication bath. The composite GPSBI was then purified by centrifugation at 5000 rpm for 15 min. Washing was performed three times with water and the composite was dried in a freeze dryer. The obtained fine powder of GPSBI was re-dispersed in DI water for further characterization.

## 3. Result and Discussion

In situ polymerizations of SB monomer on the graphene surface were carried out, and the graphite was exfoliated into thin layers with the smaller size of the graphene sheets. To know the molecular weight of PSB on the graphene surface, PSB was prepared by adopting the same procedure as GPSB and GPSBI by excluding graphene powder. The prepared PSB was characterized using size-exclusion chromatography (SEC) and TGA. The molecular number of PSB was measured as 24,536 g/mol using SEC (refer to supporting information). The GPSB and GPSBI composites were characterized using various physicochemical techniques. The composites were compared with G and IONPs.

X-ray diffraction (XRD) patterns have been utilized to study the defect polymer-stabilized graphene sheets in composites. The XRD patterns obtained for G, IONPs, GPSB, and GPSBI can be seen in Figure 1a. In G, the typical graphitic peaks at 2θ = 26.4° with d-spacing of 3.34 Å (Bragg law: d = n × λ/2θ; n = 1, λ = 0.154 nm) and at 2θ = ~55° correspond to (002) and (004) planes, respectively [45,46]. The diffraction peaks of IONPs observed at 30.33°, 35.77°, 43.43°, 53.81°, 57.44°, and 62.98° correspond to the planes (220), (311), (222), (400), (422), (511), and (440), respectively [47,48,49,50]. The size (Scherer formula: D = k × λ/(β × cosθ); k = 0.9, λ = 0.154 nm, β = full width half maximum) of IONPs was calculated as ~17.0 nm using the (311) plane [51]. The diffraction peak in GPSB was observed at 26.64°, corresponding to the (002) plane. The GPSBI composite shows peaks at 26.67°, 30.33°, 35.77°, 43.43°, 53.81°, 57.44°, and 62.98° that are attributed to (002-graphitic) and iron oxide peaks (220), (311), (222), (400), (422), (511), and (440), respectively. The interlayer distance of graphene sheets in GPSB and GPSBI were calculated as 3.36 and 3.30 Å; a slight increment in the interlayer distance suggests the partial exfoliation of graphite into graphene. Furthermore, compared with G, the diffraction peaks of GPSB and GPSBI show a slight shift and decrement in intensity. This might be due to defects in graphene sheets because of the presence of PSB and IONPs. However, the interlayer distance in GPSBI is higher than G but lower than the interlayer distance in GPSB; this can be explained by considering the presence of IONPs on graphene in addition to PSB.

Figure 1b presents the Raman spectra of G, GPSB, and GPSBI, showing three strong peaks at ~1350, ~1580, and ~2700 cm^−1^ corresponding to D, G, and 2D bands, respectively. The D band represents the defect sites in the graphene sheets at edges and surfaces and the size of the graphitic crystals [52,53]. The G band arises from the *sp*^2^ carbon–carbon bond from the first-order scattering of the E_2g_ phonon [54]. I_D_/I_G_ ratios of G, GPSB, and GPSBI were calculated as 0.13, 0.15, and 0.29, respectively. The I_D_/I_G_ ratio represents the degree of disorder and inversely relates to the size of graphene sheets [55]. The I_D_/I_G_ ratio result suggests that the size of graphene is decreasing in the order of G, GPSB, and GPSBI. In addition, the disorder in the composite increases, which confirms the functionalization of the graphene surface with PSB in GPSB and with PSB and IONPs in GPSBI. The 2D band around 2700 cm^−1^ indicates the layer of graphene sheets in the composites [56,57]. Compared to G, 2D bands in GPSB and GPSBI are sharp, indicating the thin-layered graphene sheets compared to the graphene sheets in G. Additionally, GPSBI has two sharp, distinct peaks and two small, broad peaks at ~210, ~277, ~380, and ~580 cm^−1^, attributed to the IONPs, confirmed by the Raman spectra of IONPs shown in Appendix A.

The surface morphology of the G, IONPs, GPSB, and GPSBI was examined by FESEM and TEM measurements. The FESEM images of G and corresponding elemental mapping are shown in Appendix A. The FESEM image of graphite flakes reveals graphene sheets with lateral sizes of 10 ± 4 µm. The elemental mapping measurement confirms the uniform distribution of carbon (C) elements and the fair distribution of oxygen (O) elements on the G surface. Appendix A depicts the FESEM images of IONPs, and their elemental mapping is shown in Appendix A. The even distribution of O and iron (Fe) elements is clear from the images (Appendix A).

Figure 2 displays the FESEM images (a–c) of GPSB and their elemental mapping (d–g). GPSB possesses plate-like morphology with homogenous granular size. The size of graphene sheets is much smaller than graphene sheets in G; the lateral size of graphene sheets in GPSB was measured as 3 ± 1 µm. The elemental mapping of GPSB (Figure 2b) was further analyzed to study the existing elements. The results revealed that C (Figure 2d), nitrogen (N) (Figure 2e), O (Figure 2f), and sulfur (S) (Figure 2g) are evenly distributed on the graphene sheets. The intensity of element O is significantly improved compared to element O in G. The elements (O, N, and S) originated from PSB, indicating the successful formation of composite GPSB and functionalization of graphene sheets with PSB.

FESEM images, along with the elemental mapping of GPSBI, are depicted in Figure 3. As can be seen in Figure 3, the prepared GPSBI exhibits the plate-like structure of graphene sheets with IONPs. Moreover, a large amount of IONPs are distributed homogeneously on the graphene sheets. The average lateral size of graphene sheets was measured as 3 ± 0.5 µm. The presence of elements C, N, O, and S from PSB and Fe and O from IONPs were confirmed from the elemental mapping images, as shown in Figure 3d–h.

The morphologies of G, IONPs, GPSB, and GPSBI were further investigated using TEM measurements. The TEM images of G depicted in Appendix A reveal the plate-like, wrinkled structure of graphene sheets. The lateral size of the graphene sheets in G was measured as 9 ± 4 µm.

Figure 4 exhibits the morphology of GPSB and GPSBI composites. Figure 4a–c illustrates the TEM images of GPSB composites, displaying the layered, folded, wrinkled structure with many small graphene sheets on large graphene sheets. The lateral size of graphene sheets in GPSB was measured as 2.5 ± 0.2 µm. Appendix A exposes the aggregated spherical shape IONPs particles with an average size of about 20 ± 8 nm. The size of IONPs measured from XRD (~17 nm) and TEM (20 ± 8 nm) shows a similar result. Aggregated IONPs (Appendix A) suggest that the surface is not well stabilized with the stabilizing molecules/benzyl alcohol used during the preparation of IONPs. However, the IONPs in GPSBI composites (Figure 4d) show a uniform distribution. In addition, Figure 4d–f reveals the thin graphene sheets compared to graphene sheets in GPSB (Figure 4a–c). This reveals that PSB and IONPs have effective interaction with graphene sheets in GPSBI. Thus, the uniform distribution of IONPs on the graphene surface is observed in the TEM image. The lateral size of graphene sheets in GPSBI was measured at 1.5 ± 0.2 µm, and IONPs were measured at 18 ± 5 nm, showing that the size of the IONPs is maintained in the stabilization of the graphene surface.

The XPS data provide further information about the chemical composition and formation of GPSB and GPSBI. The survey XPS spectrum of G is depicted in Appendix A, with peaks at binding energy of ~286 and ~530 eV attributed to C 1s and O 1s, respectively (Appendix A). The C 1s level deconvoluted into four peaks at 284.76, 285.52, 286.22, and 286.92 eV, attributed to C=C/C-C, C-O, C=O, and O=C-OH, respectively (Appendix A). The deconvoluted XPS spectra of O 1s for G shown in Appendix A show oxygen in different functional groups: C=O (531.80 eV), C-O (533.16 eV), and O=C-O (533.84 eV). Appendix A shows the survey spectrum of IONPs, which contains Fe and O elements. Appendix A displays the Fe 2p high-resolution spectrum, with two prominent peaks at ~711 and 724 eV assigned to Fe 2p_3/2_ and Fe 2p_1/2_, respectively, consistent with the reported IONPs [58,59]. Deconvolution of Fe 2p_3/2_ shows three peaks at 710.98, 714.54, and 719.01 eV attributed to Fe^3+^, Fe^2+^, and satellite peaks of Fe 2p_3/2_, respectively. Three peaks observed on deconvolution of Fe 2p_1/2_ at 724.08, 727.13, and 732.36 eV correspond to Fe^3+^, Fe^2+^, and satellite Fe peak of Fe 2p_1/2_, respectively. This result reveals that the major component of IONPs is Fe^3+^ with a minor component of Fe^2+^.

The XPS spectra of GPSB are shown in Figure 5, and the survey spectra of GPSB are depicted in Figure 5a. The survey spectrum indicates the presence of elements such as S 2p, C 1s, N 1s, and O 1s at 170, 285, 403, and 530 eV, respectively. On deconvolution of C 1s (Figure 5b), five peaks were observed at 284.76, 285.59, 286.37, 287.13, and 289.12 eV, responsible for C=C/C-C, C-N/C-SO_3_, C-O, C=O, and O=C-O/O-C-SO_3_, respectively. The SO_3_ arises from the sulfonate group in PSB [60,61]. The deconvoluted XPS spectra of N 1s for GPSB, shown in Figure 5c, reveal oxygen atoms in different functional groups: −N− (399.75 eV) and −N(CH_3_)_3_^+^ (402.65 eV) [60]. The O 1s level (Figure 5d) shows four peaks at 531.28, 532.40, 533.62, and 534.45 eV corresponding to C=O, C-O, O=C-OH, and H-O-H (moisture), respectively. Figure 5e shows the S 2p level with deconvolution results in four peaks at 164.06, 165.22, 167.68, and 168.89 eV attributed to oxidized S, C-SO_3_^2−^ 2p_3/2_, and C-SO_3_^2−^ 2p_1/2_, respectively. The presence of −SO_3_ and −N(CH_3_)_3_^+^ confirms the functionalization of the graphene surface with PSB.

Figure 6 shows the XPS spectra of GPSBI. The survey spectra show distinct peaks of S 2p, C 1s, N 1s, O 1s, and Fe 2p at 170, 284, 401, 532, and 712 eV, respectively (Figure 6a). Five peaks were observed on deconvolution of the C 1s level (Figure 6b) at 284.80, 285.68, 286.51, 287.31, and 289.22 eV responsible for C=C/C-C, C-N/C-SO_3_, C-OH/C-O-C, C=O, and O=C-O/O=C-SO_3_, respectively. Deconvolution of the N 1s level results in two peaks at 400.14 and 402.78 eV, attributed to the presence of −N(CH_3_)_3_^+^ and −N- groups in GPSBI, respectively (Figure 6c). Four peaks at 530.56, 531.65, 532.76, and 534.06 eV were observed on the deconvolution of O 1s (Figure 6d), responsible for C=O, C-O/Fe-O, O=C-OH, and H-O-H, respectively. Figure 6e depicts the high-resolution spectra of GPSBI at the S 2p level, resulting in four peaks on deconvolution. Two trace peaks of oxidized S at 164.21 and 165.36 eV and major peaks at 167.85 and 169.06 eV are responsible for C-SO_3_^2−^ 2p_3/2_ and C-SO_3_^2−^ 2p_1/2_, respectively. At the Fe 2p level, GPSBI showed six peaks on deconvolution, which is clear from Figure 6f. Three peaks were observed at the Fe 2p_3/2_ level at 711.41, 714.46, and 719.35 eV, which are attributed to Fe^3+^, Fe^2+^, and Fe 2p_3/2_ satellite peaks, respectively. Similarly, the Fe 2p_1/2_ level showed peaks at 724.47, 727.35, and 732.81 eV, which are responsible for Fe^3+^, Fe^2+^, and Fe 2p_1/2_ satellite peaks, respectively. This result reveals that the major composition of IONPs in GPSBI is Fe^3+^. XPS studies reveal that the major component of IONPs is Fe_3_O_4_, and it has a minor component, α-Fe_2_O_3_. Moreover, the structure of IONPS remains the same in GPSB, and GPSBI structural changes were not observed.

The thermal behaviors of G, IONPs, PSB, and prepared composites were studied by TGA. G, IONPs, GPSB, GPSBI, and PSB samples were subjected to thermal decomposition up to 900 °C in the inert atmosphere, as shown in Figure 7. The thermal pattern of G and IONPs clearly show residue of about 98.19% and 95.01%, respectively. This reveals the stability of G and IONPs. The 5% loss in IONPs and 2% loss in G might be responsible for the degradation of adsorbed water molecules or moisture on the surface. PSB showed three-step degradation, with weight loss of 5% below 200 °C, weight loss of 16.5% between 230 and 320 °C, and weight loss of 84.5% above 350 until 450 °C, which are attributed to adsorbed water molecules, the degradation of nitrogen atoms from quaternary ammonium salt, and sulfate groups, respectively [62,63,64]. The remaining polymer degraded and showed a residual weight of about 5.66% at ~900 °C. The composites GPSB and GPSBI showed three-step degradation. The 5% weight loss observed for GPSB around 100 °C was due to the degradation of adsorbed water molecules. As a second step, polymer degradation takes place with a weight loss of 23.58% between 250 and 320 °C (nitrogen groups). The sulfate groups degraded between 320 and 425 °C with a weight loss of 32.6%. Furthermore, GPSB showed residual weight loss of 60.5% observed until ~900 °C, which was attributed to the degradation of left polymer structures and decomposition of most stable oxygen functionalities [65,66]. In GPSBI composites, 5% weight loss at about 100 °C was due to the degradation of water molecules that began at about 70 °C. The PSB polymer decomposition occurred in two steps at 250–320 °C (nitrogen atoms) and 350–420 °C (sulfonate) with weight loss of ~23% and ~28%, respectively. From ~450 °C to 730 °C, weight loss might be due to the degradation of the remaining part of the polymer and degradation of IONPs left with the residual weight of about 58.8% at around 900 °C [66].

The overall studies revealed that PSB interacts with graphene surfaces through ester, quaternary ammonium, and oxygen from sulfonate units. Furthermore, sulfonate units from PSB interact simultaneously with IONPs and graphene surfaces. These interactions help to thoroughly disperse GPSB and GPSBI composites in water. Thus, the hydrophobic nature of the graphene surface can be converted to hydrophilic nature and can be applied to bio-applications.

## 4. Conclusions

We successfully prepared GPSB and GPSBI composites using a simple method. XRD and Raman studies confirm the partial exfoliation of graphite into thin-layered graphene sheets in GPSB and GPSBI composites. TEM and SEM measurements revealed the size of graphene sheets in GPSB and GPSBI as 3 ± 0.2 µm. These graphene sheets are smaller than the graphene sheets that were in G. From XRD and TEM studies, IONP size was measured as ~20 nm. The elemental mapping further showed the uniform distribution of PSB and IONPs on graphene sheets. XPS and TGA revealed the presence of PSB and IONPs in the prepared composites. TGA showed the residual weight as 60.5% and 58.8% for GPSB and GPSBI, respectively. This shows the amount of graphene that exists in the prepared composites. The combination of PSB with graphene surface showed remarkable structural and physicochemical properties with IONPs. These composites with outstanding functionalities can be potential candidates for biomedicine applications.

## Data Availability

Not applicable.

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
