# Peer review of "Synthesis of Water-Dispersed Sulfobetaine Methacrylate–Iron Oxide Nanoparticle-Coated Graphene Composite by Free Radical Polymerization"

_polymers, 2022, doi:10.3390/polym14183885_

Round 1

Reviewer 1 Report

The synthesis of composites consisting of graphene, polysulfobetaine (PSB) and ion oxide nanoparticles (IONPs) was described in this manuscript. It is a very interesting idea and the products are definitely promising for numerous applications. The work is well described but from the side of a material scientist and a polymer physicist. Basic knowledge is nor provided in this work.

·         There is no idea about the molecular weight of the PSB and its dispersity and how these parameters may affect the properties of the composites.

·         What is the amount of polymer-IONPs-graphene in the composites and how the properties of the final materials change upon changing the composition.

·         Is it possible to prepare the composites by first polymerizing the zwitterionic monomer and then interact the polymer with the graphene?

·         Do we have chemical attachment of the polymer to the graphene sheets or adsorption only takes place?    

·         What is the source of graphene used in this study? It was pure graphene or it was subjected to any kind of treatment?

·         The water from the surface is not decomposed but just evaporated upon heating.  

Author Response

We appreciate the time and efforts spent by the referee in reviewing our manuscript. We sincerely thank the reviewers for the valuable comments and suggestions that helped to improve the quality of the manuscript. We have addressed all the issues pointed out by the reviewers in the revised manuscript.

Comments from the referees:
Reviewer #1: The synthesis of composites consisting of graphene, polysulfobetaine (PSB) and ion oxide nanoparticles (IONPs) was described in this manuscript. It is a very interesting idea and the products are definitely promising for numerous applications. The work is well described but from the side of a material scientist and a polymer physicist. Basic knowledge is nor provided in this work.

Response: Thank you very much for the positive comments and suggestions. The manuscript has been revised point by point to improve the quality. Kindly note the title has been revised. The properties of the composite were not studied using the changing ratio of graphene, IONPs, or PSB. Just the properties of the prepared composites were studied.

  • There is no idea about the molecular weight of the PSB and its dispersity and how these parameters may affect the properties of the composites.

Response: Thank you very much for your comments. Kindly note that the SB monomer was polymerized in situ on the graphene surface. So it is difficult to find the molecular weight of PSB that interacts with the graphene surface. Also, the ratio of monomer with IONPs and graphene was not varied, and the properties of PSB with different molecular weight was not studied. The authors aim to show that stable water-dispersible graphene composites can be prepared by a simple method.

  • What is the amount of polymer-IONPs-graphene in the composites and how the properties of the final materials change upon changing the composition.

Response: Thank you very much for your perceptive comment. Kindly note that the amount of polymer-IONPs-graphene in the composites was calculated using TGA measurement in weight % as graphene-98.2; IONPs-95.2; GPSB-61; GPSBI-59; and PSB-5.66.

  • Is it possible to prepare the composites by first polymerizing the zwitterionic monomer and then interact the polymer with the graphene?

Response: Thank you very much for the insightful comments. We published a few articles (J. Colloid Interface Sci. 497, 2017, 359-367 and J. Colloid Interface Sci. 464, 2016, 25-35) where polymers (polyvinyl pyridine-, polystyrene-, and poly pyrene-based block copolymers with polyethylene glycol) were prepared and after characterization of polymers, graphene composites were prepared. Recently we published a work (Int. J. Hydrog. Energy, 46, 2021, 10850-10861), in which 2-(methacryloyloxy)ethyl phosphorylcholine and poly (ethylene glycol) monomethacrylate (PMPC-co-PEG) polymer was prepared first, and the graphene composites were prepared. Also, by in situ polymerizations of MPC and PEG monomers, the graphene surface was stabilized with PMPC-co-PEG polymer. Graphene composite prepared by in situ polymerization showed stable graphene composite dispersion among these composites. Thus, the present manuscript (Polymers) in situ polymerization of SB on graphene surfaces was adopted.  

  • Do we have chemical attachment of the polymer to the graphene sheets or adsorption only takes place?    

Response: Thank you very much for your comment. The composites are prepared by physical treatment (sonication), so PSB is adsorbed on the graphene surface. In the revised manuscript, the interaction of PSB and IONPs with graphene surface has been discussed at the end of the result and discussion part. Kindly refer to page 17.

  • What is the source of graphene used in this study? It was pure graphene or it was subjected to any kind of treatment?

Response: Graphite powder purchased from Aldrich was used directly without any treatment.

  • The water from the surface is not decomposed but just evaporated upon heating.

Response: Thank you very much for the comments. The line has been revised as 5% weight loss at about 100 degrees due to the degradation of water molecules that began at about 70 degrees. Kindly refer to page 17. 

Reviewer 2 Report

In this manuscript, water soluble sulfobetaine methacrylate-iron oxide nanoparticles coated graphene composite (GPSB I) and sulfobetaine methacrylate coated graphene surface (GPSB ) were prepared by free radical polymerization. The structures of GPSB and GPSBI was well confirmed by X-ray diffraction, Raman, field emission scanning electron microscopy, transmission electron microscopy, X-ray photoelectron spectroscopy, and thermogravimetric analysis. It can be recommended for publication after possible revision.

1. What is the structure of IONPs? and is there any new properties after the introduction of IONPs?

2. There is a couple of errors in the manuscript, the authors should pay attention to  the grammar and English writing.

Author Response

We appreciate the time and efforts spent by the referees in reviewing our manuscript. We sincerely thank the reviewers for the valuable comments and suggestions that helped improve the quality of the manuscript. We have addressed all the issues pointed out by the reviewers in the revised manuscript.

Comments from the referees:

Reviewer #2: In this manuscript, water soluble sulfobetaine methacrylate-iron oxide nanoparticles coated graphene composite (GPSB I) and sulfobetaine methacrylate coated graphene surface (GPSB ) were prepared by free radical polymerization. The structures of GPSB and GPSBI was well confirmed by X-ray diffraction, Raman, field emission scanning electron microscopy, transmission electron microscopy, X-ray photoelectron spectroscopy, and thermogravimetric analysis. It can be recommended for publication after possible revision.

Response: Thank you very much for your positive comments and suggestions. The manuscript has been revised point by point to improve its quality.

  1. What is the structure of IONPs? and is there any new properties after the introduction of IONPs?

Response: Thank you very much for the insightful comment. As per the XPS studies, the structure of IONPs is a mixture of magnetite (Fe3O4) and hematite (α-Fe2O3). However, among these major components will be the Fe3O4 structure. The sentences about the structural details of IONPs in the revised manuscript have been included. Kindly refer to page 15.

  1. There is a couple of errors in the manuscript, the authors should pay attention to  the grammar and English writing.

Response: Thank you very much for your comment. The manuscript has been thoroughly checked to avoid grammar and English mistakes.

Round 2

Reviewer 1 Report

The authors have considerably increased the quality of their manuscript. However, there is need to have at least an idea of the molecular weight of the adsorbed chains on the graphene surface. The authors could repeat experiments without the presence of graphene and measure the molecular weight of their polymers and also it would be helpful to provide the stoichiometric molecular weights for comparison. After that the work can be published in Polymers. 

Author Response

Comments from the referees:
Reviewer #1: The authors have considerably increased the quality of their manuscript. However, there is need to have at least an idea of the molecular weight of the adsorbed chains on the graphene surface. The authors could repeat experiments without the presence of graphene and measure the molecular weight of their polymers and also it would be helpful to provide the stoichiometric molecular weights for comparison. After that the work can be published in Polymers. 

Response: Thank you very much for the positive comments and suggestions. As suggested PSB was characterized using size-exclusion chromatography to know the molecular weight. The Mn value of PSB was measured as 24536 g/mol, PD = 1.8. Kindly refer to pages 6 in the manuscript and page 2 in the supporting information.
